# Development of Electromagnetic-Wave-Shielding Polyvinylidene Fluoride–Ti_3_C_2_T_x_ MXene–Carbon Nanotube Composites by Improving Impedance Matching and Conductivity

**DOI:** 10.3390/nano13030417

**Published:** 2023-01-19

**Authors:** Qimei Zhang, Jian Cui, Shuai Zhao, Guangfa Zhang, Ailin Gao, Yehai Yan

**Affiliations:** 1Key Laboratory of Rubber-Plastics, Ministry of Education/Shandong Provincial Key Laboratory of Rubber-Plastics, School of Polymer Science and Engineering, Qingdao University of Science and Technology, Qingdao 266042, China; 2School of Materials and Environmental Engineering, Chizhou University, Chizhou 247000, China

**Keywords:** electromagnetic interference shielding, segregated structure, polyvinylidene fluoride, MXene, single-walled carbon nanotubes

## Abstract

Absorption-dominated electromagnetic interference (EMI) shielding is attained by improving impedance matching and conductivity through structural design. Polyvinylidene fluoride (PVDF)–Ti_3_C_2_T_x_ MXene–single-walled carbon nanotubes (SWCNTs) composites with layered heterogeneous conductive fillers and segregated structures were prepared through electrostatic flocculation and hot pressing of the PVDF composite microsphere-coated MXene and SWCNTs in a layer-by-layer fashion. Results suggest that the heterogeneous fillers improve impedance matching and layered coating, and hot compression allows the MXene and SWCNTs to form a continuous conducting network at the PVDF interface, thereby conferring excellent conductivity to the composite. The PVDF-MXene-SWCNTs composite showed a conductivity of 2.75 S cm^−1^ at 2.5% MXene and 1% SWCNTs. The EMI shielding efficiency (SE) and contribution from absorption loss to the total EMI SE of PVDF-MXene-SWCNTs were 46.1 dB and 85.7%, respectively. Furthermore, the PVDF-MXene-SWCNTs composite exhibited excellent dielectric losses and impedance matching. Therefore, the layered heteroconductive fillers in a segregated structure optimize impedance matching, provide excellent conductivity, and improve absorption-dominated electromagnetic shielding.

## 1. Introduction

With the rapid advancement of network and information technology, electronic devices have been popularized in facilitating everyday communication. This also inevitably introduce issues such as electromagnetic interference (EMI), information leakage, and radiation hazards [1,2]. Thus, electromagnetic (EM) shielding materials that effectively protect the human body and precision equipment from radiation interference have attracted considerable attention [3]. With their advantages of low weight, easy processing, and superior specific strength, conductive polymer composites (CPCs) are known to play a vital role in promoting the EMI shielding features of microwave absorption [4,5]. In particular, two-dimensional (2D) transition metal carbides, nitrides, and carbonitrides (MXenes) are widely used to fabricate high-efficiency EMI shielding composites and have become some of the most promising electronic materials. The exceptional conductivity of MXenes provides an inherent advantage by extending the dissipative path of incident EM waves, thereby improving the efficacy of electromagnetic shielding [6]. Moreover, the EM shielding performance of MXenes is enhanced because fine-tuned impedance matching can be performed [7], which endows the Ti_3_C_2_T_x_ MXene/polymer composite with considerable market potential [8]. The absorption-dominated efficient EMI shielding properties of MXene composites have been investigated by numerous researchers. By electrostatically assembling negative MXene sheets onto positive polystyrene microspheres, Sun et al. [9] fabricated highly conductive MXene/polystyrene nanocomposites. After compression, the composite exhibited an excellent EMI shielding performance of more than 54 dB across the entire X-band, with a low percolation threshold of 0.26 vol%.

However, the excessive accumulation of MXene layers results in a sharp reduction of their electrical conductivity [10], thus limiting its application in electromagnetic shielding polymer composites. Single-walled carbon nanotubes (SWCNTs) with excellent conductivity and stable mechanical properties can be welded to the 2D MXenes layers to perform a connecting and supporting role [11], resulting in excellent EMI shielding [12]. Cao et al. [13] used simple alternating vacuum-assisted filtration technology to construct an ultrathin and flexible carbon nanotube/MXene/cellulose nanofiber composite paper, a gradient and sandwich-structured paper, which exhibited a high conductivity of 2506.6 S m^−1^ and a high EMI shielding effectiveness of 38.4 dB. Weng et al. [14] reported the synthesis of a multifunctional and translucent MXene/carbon nanotube sandwich structure assembly composite film using the layer-by-layer spin spray method. A strong specific shielding of 58187 dB cm^−2^ g^−1^ was achieved, along with a conductivity of up to 130 S cm^−1^.

The excellent conductivity of SWCNTs endows the composite with a high permittivity [15], which leads to impedance mismatch at the composite interface [16,17]. Simultaneously, SWCNTs composites significantly reflect the incident electromagnetic waves on their surface, resulting in secondary EM radiation contamination. Jia et al. [18] developed a segregated structure polymer composite (SPCs) consisting of carbon nanotube (CNT)/polyethylene. The electromagnetic shielding absorption coefficient significantly decreased as the CNT content increased, while the conductivity of the composite increased. The EM wave absorption performance of a shield depends not only on efficient EM wave attenuation but also on impedance matching [19,20].

Traditional loss materials exhibit inferior absorption properties of EM waves due to poor impedance matching. Efficient EMI materials with strong absorption have been widely recognized [21]. Therefore, the optimization of impedance matching is essential.

Optimizing the impedance matching of a composite and abundant heterostructure interphase and increasing the conductivity improve EM shielding and absorption performance [18]. However, in the segregated structure, the method of optimizing impedance matching by mixing different kinds of fillers deteriorates the conductivity [22]. Park et al. [23] proposed that decreased conductivity can be attributed to the disruption of CNT networks. The variation in the conductivity of CNTs composites containing secondary fillers depends on the filler size [24].

Thus, a rational route to increase the absorption-dominated EM shielding in CPCs by optimizing impedance matching and improving conductivity can be explored using layered heterostructures. At present, the development of lightweight, practical CPCs with enhanced conductivity remains a challenge. An effective strategy for solving these problems is by controlling the distribution of the conductive fillers by SPCs. Due to their low filling content and high conductivity, ultra-high shielding properties can be achieved in SPCs [25]. Additionally, the lower processing temperature of the segregated structure avoids damage to the structure of the MXene [26]. PVDF is selected as a composite polymeric matrix of inorganic–organic composites owing to its excellent dielectric properties [27], superior mechanical properties [28], and tailored shape [29].

In this paper, we propose a novel strategy for improving conductivity and optimizing impedance matching by layered hetero-filling (MXene and SWCNTs) of a PVDF-MXene-SWCNTs segregated structural composite. The PVDF-MXene-SWCNTs segregated structural composite is fabricated by the electrostatic flocculation process and compression molding. SWCNTs and MXene are layered independently in the segregated PVDF composites optimized for impedance matching, leading to superior absorption coefficients. The layered fillings of SWCNTs and MXene in segregated PVDF composites allows their high conductivity to be preserved, resulting in excellent dielectric losses. The synergistic effect of a completely conducting network with a layered segregated structure consisting of layered SWCNTs and an MXene leads to excellent absorption-dominated EM shielding.

## 2. Materials and Methods

### 2.1. Materials

Titanium aluminum carbide powder (Ti_3_AlC_2_, MAX phase, 99%) was purchased from Shanghai Bowei Applied Materials Technology Co., Ltd., Shanghai, China. Polyvinylidene fluoride (PVDF) was purchased from Zhejiang Fluorine New Chemical Materials Co., Ltd., Hangzhou, China. Polyethyleneimine (PEI) was purchased from Shanghai McLean Biochemical Technology Co., Ltd., Shanghai, China. We purchased sodium dodecyl benzene sulfonate (SDBS) from Sinopharm Chemical Reagent Co., Ltd., Shanghai, China, and lithium fluoride (LiF) from Shanghai Kangtuo Chemical Co., Ltd., Shanghai, China. SWCNTs (purity: >95 wt%, diameter: 1–2 nm, length: 5–30 µm) were obtained from the Chengdu Organic Chemistry Co., Ltd., Chengdu, China. All reagents were not purified.

### 2.2. Preparation

Synthesis of Ti_3_C_2_T_x_ (MXene): MXene was prepared by etching MAX using LiF based on the literature [30]. Hydrochloric acid aqueous solution (20 mL, 6 mol L^−1^) was added to 1.6 g LiF powder at 45 °C. After stirring for 10 min, 1.0 g MAX powder was added slowly, and the mixture was stirred continuously for 48 h. Thereafter, 100 mL hydrochloric acid (1 mol L^−1^) was added to the etched MAX. The sediment was washed and centrifuged until a pH of 6–7 was achieved. The etched MAX was exfoliated by ultrasonication (100 W) in ice baths and nitrogen atmosphere for 30 min, after which the desired MXene solution was obtained after centrifugation.

The preparation of dispersion of SWCNTs: SWCNTs (100 mg) were added to SDBS aqueous solution (250 mL, 0.1 mg mL^−1^), followed by treatment with an ultrasonic cell pulverizer (300 W) for 1 h. Uniformly dispersed SWCNTs were obtained.

Preparation of PVDF/MXene microspheres and composite: PVDF treated with PEI was added to the MXene dispersed solution and magnetically stirred to flocculate. The PVDF/MXene microspheres were fabricated after filtration, washing, and drying. PVDF/MXene microspheres with 0.05%, 0.5%, 1%, 2.5%, 4%, and 5% MXene were prepared successively and labeled as PVDF/MXeneX (X = the mass fraction of MXene). The PVDF/MXene composite was prepared by hot pressing the microspheres under the conditions of 170 °C and 12 MPa for 20 min and subsequent cooling to 25 °C ± 5 °C. A sample with thickness of 2.2 mm ± 0.3 mm was achieved by controlling the quality of raw materials.

Preparation of PVDF/SWCNTs microspheres and composites: PVDF/SWCNTs microspheres with 0.01%, 0.2%, 1%, 1.5%, 2%, and 2.5% SWCNTs were prepared similarly to PVDF/MXene microspheres. The microspheres were labeled as PVDF/SWCNTsX (X is the mass fraction of SWCNTs). The PVDF/SWCNTs composite was prepared following the molding procedure described above.

Preparation of PVDF-MXene-SWCNTs composite: Two grams of PVDF/MXene (MXene percent is 2.5 wt%) microspheres was treated by PEI again and then added into a disperse SWCNTs in SDBS aqueous solution (100 mL and 0.1 mg mL^−1^). The PVDF-MXene2.5-SWCNTs1 composite was subsequently obtained by stirring, filtering, drying, and hot compression. The control PVDF/MXene2.5/SWCNTs1 and PVDF-SWCNTs1-MXene2.5 were prepared in a similar manner. The details of the procedure are given in the Appendix A. The essential difference between composites is the difference in structural units. The structural units of PVDF/MXene5, PVDF/SWCNTs2, PVDF/MXene/SWCNTs, PVDF-SWCNTs-MXene, and PVDF-MXene-SWCNTs composite are schematically illustrated in Figure 1.

The preparation of PVDF/MXene5, PVDF/SWCNTs2, PVDF/MXene/SWCNTs, PVDF-SWCNTs-MXene, and PVDF-MXene-SWCNTs composite by electrostatic flocculation is schematically illustrated in Appendix A. Photographs of freshly prepared PVDF/MXene5, PVDF/SWCNTs2, PVDF-SWCNTs-MXene, and PVDF-MXene-SWCNTs composite microspheres and the PVDF/MXene2.5/SWCNTs1 composite sample are shown in Appendix A. The weights by percentage for preparing the composites are listed in Table 1.

### 2.3. Characterization

A Malvern ZS90 analyzer (United Kingdom) was used to perform the zeta potential analysis. A scanning electron microscope, named Phenom, was used to characterize the cross-section of the cut and break of the composite, which was acquired from Phenom Scientific Instruments (Shanghai) Co., Ltd., Netherlands. An X-ray diffractometer (DX-2700), purchased from Dandong Haoyuan Instrument Co., Ltd. Dandong, China, was used with a Cu Kα target under radiation with a wavelength of 0.154 nm over the diffraction angle (2*θ*) range of 5–90°, with a scanning rate of 2° min^−1^. Prior to the transmission electron microscopy (TEM, Tecnai G2 F20, from FEI, Hillsboro, OR, USA) analysis, the MXene supernatant was exfoliated after centrifugation, diluted with deionized water, and added dropwise onto a Cu mesh after ultrasonic dispersion. Thereafter, the sample was dried and subjected to TEM analysis. The conductivity was evaluated using a four-probe RTS-4 detector from Guangzhou Four-Probe Technology Co., Ltd., Guangzhou, China.

For the EM shielding test, the waveguide method was tested over a frequency range of 8.2–12.4 GHz in the X-band, with sample dimensions of 10.12 mm × 22.86 mm × 2.2 mm. The test was performed at 25 °C ± 5 °C using a vector network analyzer (ZNB 20, Robert & Schwartz Technologies, Inc., Munich, Germany.) The absorption loss to the total EMI SE (absorption coefficient) was determined by the following Equation (1):(1)$$\mathrm{Absorption}\;\mathrm{coefficient}=\mathrm{absorption}\;\mathrm{EM}\;\mathrm{shielding}\left({\mathrm{SE}}_{A}\right)/\mathrm{total}\;\mathrm{EM}\;\mathrm{shielding}\left({\mathrm{SE}}_{T}\right)$$

## 3. Results and Discussion

### 3.1. Characterization of MXene

The few-layer Ti_3_C_2_T_x_ MXene is obtained from the analysis in Figure 2a–f. Figure 2a and b illustrate the scanning electron microscopy (SEM) images of MXene before and after exfoliation, respectively. As shown in Figure 2b, the MXene is transparent when the electron beam of the SEM hits its surface; the image reveals the few-layer structure of MXene [31]. A TEM image of MXene (Figure 2c) shows the presence of a few layers of 2D sheet-like shapes. Compared to those in the SEM image, the nanosheets in the TEM image are narrower in size and have some imperfections, which may be related to the ultrasound processing. As shown in Figure 2d, well-organized and extremely ordered subnanometer channels are formed due to the even distribution of the termination group (-OH, -F, and -O) [32] on the surface of the MXene sheets [33,34].

The X-ray diffraction (XRD) patterns of the precursors MAX and MXene are shown in Figure 2e. The peaks at 2*θ* = 9.51°, 19.1°, 34.0°, 39.0°, 44.8°, and 60.1° in the XRD pattern for MAX correspond to its (002), (004), (101), (104), (105), and (110) crystal planes, respectively [35], where the (104) peak represents the A1 phase [36]. The XRD pattern of MXene shows that the characteristic peak at 34.0° ((104) crystal plane) disappears, indicating that the Al atoms in the MAX phase are removed by etching. Upon comparing the XRD patterns of MXene and MAX, the sharp diffraction peak of the (002) crystal plane [37] representing the interlayer spacing evidently shifts from 9.51° to 7.00°, indicating that MXene has a greater interlayer spacing. In Figure 2f, the MXene solution is transparent and green after dilution, clearly showing the Tyndall effect together with the zeta potential (Appendix A), which indicates that the MXene dispersion is stable [38]. Based on the above analysis, it can be concluded that the few-layer Ti_3_C_2_T_x_ MXenes were successfully prepared.

### 3.2. Morphology of the SWCNTs

The SWCNTs are uniformly dispersed in the SDBS solution, as seen in Figure 2g, where no clumps are visible to the naked eye after dilution. Figure 2h,i represent the SEM images of the SWCNTs before and after dispersion, respectively. In Figure 2h, the SWCNTs are evidently clustered and entangled before dispersing.

As shown in Figure 2i, the morphology of disentangled SWCNTs illustrates the successful preparation of uniformly dispersed SWCNTs, indicating that the SDBS and mechanical forces in the dispersion are significant for uniformly dispersed SWCNTs. Note that the uniform dispersion of SWCNTs is a prerequisite for the integrity and uniform coating of PVDF microspheres.

### 3.3. Potential Analysis of Electrostatic Self-Assembly

The Ti atoms on the surface of the MXene easily bind to the molecules or ions in a solution, such as H_2_O and HF, and carry numerous groups, e.g., -F and -OH [39], which enhance hydrophilicity and make the surface electronegative (−106.78 mV) (Appendix A). The PVDF microsphere aqueous dispersion treated with PEI is positively charged (22.03 mV), and the SWCNTs dispersed with SDBS exhibit a potential of −77.94 mV, which satisfies the electrostatic flocculation process by electrostatic interactions.

### 3.4. SEM Analysis of Composite Materials

The SEM images were used to confirm the layered fillers of the composite material with segregated structures before and after hot pressing. Figure 3(a1) shows the SEM image of the PVDF/MXene microspheres, indicating that the MXene is completely and uniformly coated on the surface of the PVDF microsphere due to electrostatic effects. Figure 3(a2,a3) show the cross-sectional morphology and enlarged view of the PVDF/MXene composite. Evidently, MXene is uniformly distributed around the deformed PVDF interface, thus forming a continuous conducting network after hot pressing.

Figure 3(b1,b2) are SEM images depicting enlarged views of the composite microspheres of PVDF/SWCNTs after electrostatic flocculation processes. Figure 3(b2) illustrates a uniform coating of SWCNTs on a PVDF microsphere. A continuous conducting network of SWCNTs wrapping around the PVDF can also be clearly observed in the PVDF/SWCNTs composite (Figure 3(b3)). Therefore, the electrostatic flocculation process ensures that MXene (Figure 3(a1)) and SWCNTs (Figure 3(b2)) are evenly coated on the entire surface of PVDF microspheres, thereby laying a structural foundation for the excellent conductivity of the composite materials.

Figure 3(c1) shows the profile morphology obtained by cutting. The segregated structure has a clear profile, and the fillers (MXene and SWCNTs) are evenly distributed. From the cross-sectional view of PVDF/MXene2.5/SWCNTs1 shown in Figure 3(c2) and the enlarged SEM image in Figure 3(c3), some PDVF particles are covered by MXene, while others are wrapped by SWCNTs. The surface of the deformed PVDF microsphere is coated with either MXene or SWCNTs. For the composite, fillings of MXene or SWCNTs are excluded from the deformed PVDF by squeezing; a bilayer heterogeneous filler is formed between the PVDF microspheres and the MXene and SWCNTs overlapping with each other, which are arranged in an orderly manner and form a cell-like structure [39,40] and a conductive network. Consequently, the bilayer hetero-conducting packings create bilayer shielding surfaces.

As shown in Figure 3(d1) and Appendix A, MXene covers the PVDF microspheres wrapped by the SWCNTs, which cling to the PVDF microspheres after molding. In Figure 3(d2) and the enlarged SEM image in Figure 3(d3), MXene and SWCNTs are present at the PVDF interface independent of each other. The MXenes are squeezed in the middle of SWCNTs. Because the conductivity of PVDF/MXene is inferior to that of the PVDF/SWCNTs composite, MXene may form a barrier in the conductance path of PVDF-SWCNTs-MXene2.5. We note that SWCNTs and MXene form layers around the PDVF particles, creating a four-layer shield between the PDVF particles.

In Figure 3(e1) and Appendix A, the MXene and SWCNTs at the PVDF interface are also independent. SWCNTs are overlaid on a PVDF microsphere encapsulated by MXene. After molding, the SWCNTs are directly in contact with the outer layer of the PVDF-MXene2.5-SWCNTs1 structural unit (Figure 3(e2,e3)). Its excellent conductivity is ensured by a connected network of conductive paths, which is a structural guarantee for high conductivity and high-efficiency EM shielding.

### 3.5. Optimization of the Filling Content in Composites

The conductivity of PVDF/MXene and PVDF/SWCNTs increases with increasing concentrations of MXene and SWCNT, respectively. In Figure 4a, the conductivity of the PVDF/MXene composite increases rapidly with increasing MXene concentration. For 0.5 wt% MXene, the conductivity of the PVDF/MXene composites increases to 2.24 × 10^−5^ S cm^−1^, and for 5 wt% MXene, the conductivity of PVDF/MXene reaches 0.771 S cm^−1^, which benefits from the electrostatic uniform coating and segregated structure. As shown in Figure 4b, the conductivity of the PVDF/SWCNTs composite increases significantly with increasing SWCNT content. Surprisingly, the conductivity increases to 2.14 S cm^−1^ at 2% concentration, after which the conductivity of the composite stabilizes with increasing SWCNT content. The conductivity of PVDF/SWCNTs composites is significantly higher than that of PVDF/MXene. This higher conductivity is mainly because the longer SWCNTs [41] are dispersed uniformly and can be more easily and continuously arranged and overlapped in segregated structures exhibiting a higher conductivity [42].

Conductivity is known to play a crucial role in determining the EMI shielding performance [43,44]. The conductivity values of PVDF/MXene and PVDF/SWCNTs are consistently too low to satisfy EMI shielding-related applications when the addition of MXene is less than 2.5 wt% and the PVDF/SWCNTs is less than 0.2 wt. However, electrical conductivity is not the absolute criterion for EMI shielding [42,45], especially for the absorption-dominated EM shielding performance. Therefore, the MXene and SWCNT concentrations in the composite are optimized, as listed in Table 2.

As shown in Table 2, the conductivity, EMI SE_A_, and EMI SE_T_ increase with the SWCNTs content. The conductivity and EMI SE_T_ of PVDF/MXene2.5/SWCNTs1.5 increased up to 3.77 S cm^−1^ and 45.9 dB, respectively. However, the absorption coefficient (SE_A_/SE_T_) reached a maximum of 73.3% when the content of SWCNTs was 1%. An excess of SWCNTs does not favor an increase in the absorption coefficient. To obtain efficient absorption-dominated EMI shielding, 2.5% MXene and 1% SWCNTs were employed in the composite material used in this study.

### 3.6. Conductivity, EMI Shielding, Impedance Matching, and Absorption Coefficient

To further improve impedance matching and conductivity, differences in conductivity and EMI shielding between PVDF/MXene2.5/SWCNTs1, PVDF-SWCNTs1-MXene2.5, and PVDF-MXene2.5-SWCNTs1 were also detected.

As shown in Figure 5a, PVDF-MXene2.5-SWCNTs1 (2.75 S cm^−1^) has a clear advantage over PVDF/MXene2.5/SWCNTs1 (1.05 S cm^−1^) and PVDF-SWCNTs1-MXene2.5 (0.97 S cm^−1^) in conductivity using the same fillers. This sharp distinction is due to the structural units of the composites listed in Table 1. PVDF-MXene2.5-SWCNTs1 has a barrier-free, complete, continuous conducting network from overlapping SWCNTs.

In addition, MXene increases the density of SWCNTs in layered fillings. Consequently, the conductivity (2.14 S cm^−1^) of PVDF-MXene2.5-SWCNTs1, which exceeded PVDF/SWCNTs, was at the maximum. In summary, the excellent conductivity of the PVDF-MXene2.5-SWCNTs1 composite is due to the continuous and complete conductive network formed by the SWCNTs in the layered segregated structure. However, the MXene in the interlayer of SWCNTs becomes an obstacle to the conductance path, which can be inferred from Figure 4 and Figure 5a. Therefore, the conductivity of PVDF-SWCNTs1-MXene2.5 was only 0.97 S cm^−1^. The conductivity of PVDF/MXene2.5/SWCNTs1 (1.05 S cm^−1^) is intermediate and between those of PVDF/MXene5 (0.771 S cm^−1^) and PVDF/SWCNTs2 (2.14 S cm^−1^), indicating that conductivity is still maintained by the conducting network in the composite with SWCNTs in which MXene is mixed, squeezed, overlapped, and connected. 

Figure 5b shows the EMI SE_T_ of the composite, which exhibits a different trend from that of the conductivity. The EMI SE_T_ of the PVDF/MXene2.5/SWCNTs1 composite is 41.2 dB, which is significantly higher than those of PVDF/SWCNTs2 (33.9 dB) and PVDF/MXene5 (9.94 dB). However, the EMI SE_T_ of PVDF-SWCNTs1-MXene2.5 is only 35.7 dB owing to the lower conductivity. The EM shielding of PVDF/MXene/SWCNTs composite results from the synergistic effect of MXene and SWCNTs [14]. 

Figure 5c shows the SE_R_ of the composite. The maximum average EM SE_R_ (13.1 dB) is observed with PVDF/SWCNTs. The average SE_R_ of PVDF/MXene2.5/SWCNTs1 is only 10.5 dB, while that of PVDF-MXene2.5-SWCNTs1 is only 6.56 dB, which is the minimum value excluding PVDF/MXene5. The EM SE_R_ accounts for an essential part of EM shielding, which is attributed to poor impedance matching with the air inside the CNTs [16]. The high conductivity (2.75 S cm^−1^) (Figure 5a) and low average SE_R_ (6.56 dB) (Figure 5c) of PVDF-MXene2.5-SWCNTs1 are also attributed to the optimized impedance matching.

The impedance matching of the composite is introduced to help understand the conflict between the unexceptional conductivity and prominent EM shielding the composite. The impedance matching ratios can be calculated using Equation (2) [46]: (2)$$Z_{r}=\left|\frac{Z_{\mathrm{in}}}{Z_{0}}\right|=\left|\sqrt{\mu_{r}/\varepsilon_{r}}\tan h\left[j\left(2\pi fd/c\right)\sqrt{\mu_{r}\varepsilon_{r}}\right]\right|$$
where *Z*_r_ is the impedance matching ratio, and *Z*_in_ and *Z*_0_ are the input impedance of the shield and the impedance of free space. Here, *f*, *d*, and *c* are the frequency of the EM wave, the thickness of the sample, and the speed of light, respectively. The impedance matching ratio value is close to one, indicating that the electromagnetic wave is totally absorbed with minimal reflection loss [47]. 

As shown in Figure 5d, PVDF-MXene2.5-SWCNTs1 (0.20–0.27) exhibited the highest impedance matching ratio among all the composites except for PVDF-MXene5 (0.13–0.26), thereby revealing small reflection loss EMI shielding (6.56 dB). The superior impedance matching between the vacuum and shield resulted in an improved absorption value [48]. A much higher prominent EMI SE_T_ of 46.1 dB was observed in PVDF-MXene2.5-SWCNTs1. We noted the contrast in the conductivity and electromagnetic shielding of the composite. The conductivity of PVDF/MXene2.5/SWCNTs1 (1.05 S cm^−1^) was an intermediate value between that of PVDF/MXene5 (0.771 S cm^−1^) and PVDF/SWCNTs2 (2.14 S cm^−1^). Nevertheless, the EMI SE_T_ of PVDF/MXene2.5/SWCNTs1 was 41.2 dB, which was higher than that of PVDF/SWCNTs2 (33.9 dB) and PVDF/MXene5 (9.94 dB). The contradiction between the unexceptional conductivity and prominent EM shielding is related to the better impedance matching of PVDF/MXene2.5/SWCNTs1 (0.066–0.072) than that of PVDF/SWCNTs2 (0.031–0.039). Better impedance matching achieves a higher EMI shielding.

Figure 5e presents the EM shielding absorption coefficient of the composite. The absorption coefficient of the EM shielding of PVDF/SWCNTs was only 59.57%. The absorption loss to the total EMI SE of PVDF-MXene2.5-SWCNTs1 was an overwhelming 85.7%, which was greater than those of PVDF-SWCNTs1-MXene2.5 (64.2%) and PVDF/MXene2.5/SWCNTs1 (73.3%). This finding confirms the improvement in the impedance matching performance of the composites [35]. Heterogeneous fillers were used to improve the impedance matching performance and, consequently, the absorption of the composite EM shielding. The impedance matching performance is significantly improved. 

### 3.7. Comparison of Theoretical and Experimental Absorption Performance 

To gain more insight into the absorption properties of the EMI shield, a comparison of theoretical and experimental absorption properties was performed. The experimental SE_A_ values of PVDF/MXene5, PVDF/SWCNTs2, PVDF/MXene2.5/SWCNTs1, PVDF-SWCNTs1-MXene2.5, and PVDF-MXene2.5-SWCNTs1 are shown in Figure 6a. The EMI SE_A_ values in the composites are proportional to their EMI SE_T_ values and show no significant dependence on the frequency at 8.2–12.4 GHz.

The Schelkunoff theory [49] represents the attenuation mechanism of shields to electromagnetic waves. The electromagnetic interference total shielding efficiency (EMI SE_T_) is the sum of absorption (SE_A_), reflection (SE_R_), and multiple reflections (SE_M_) [50,51]:(3)$${\mathrm{SE}}_{T}={\mathrm{SE}}_{A}+{\mathrm{SE}}_{R}+{\mathrm{SE}}_{M}$$
(4)$${\mathrm{SE}}_{A}=0.131\;d\sqrt{f\;\mu_{r}\sigma_{r}}$$
where *t* is the thickness of the shielding material, *f* is the electromagnetic wave frequency, and *σ*_r_ and *μ*_r_ are the electrical conductivity relative to pure copper and relative magnetic permeability, respectively. The equation is valid for material with thicknesses above the skin depth [52].

According to free electric theory, conductivity (σ) in the alternating electric field can be estimated that [53,54]:(5)$$\sigma \approx 2\pi f\varepsilon_{0}\varepsilon^{\prime \prime }$$
where *f* is frequency, *ε*_0_ is the permittivity of free space, and *ε″* is the imaginary permittivity.

The electric conductivity of copper is 5.8 × 10^5^ S cm^−1^ [55]; *ε*_0_ is 8.854 × 10^−12^ F m^−1^ [56]. The magnetic permeability is minuscule and close to zero (Appendix A), and the relative permeability of the nonmagnetic material is close to one [57]. According to Equations (4) and (5), the calculated theoretical SE_A_ values at *f* = 8.2–12.4 GHz and *t* = 2.2 mm were plotted, as shown in Figure 6b.

The theoretical SE_A_ values of PVDF/MXene5, PVDF/SWCNTs2, PVDF/MXene2.5/SWCNTs1, PVDF-SWCNTs1-MXene2.5, and PVDF-MXene2.5-SWCNTs1 are shown in Figure 6b. The theoretical SE_A_ increases with frequency throughout the measured frequency region. However, the overall trend is obviously different from the experimental SE_A_. First, the theoretical SE_A_ was clearly different from the experimental SE_A_, which may be due to the assumptions in Equation (4). Second, the dependence on frequency was different. The theoretical SE_A_ was more dependent on frequency and relative electrical conductivity (*σ*_r_). However, the experimental SE_A_ was more complex, and the contribution from multiple reflections (SE_M_) cannot be ignored [58]. Thus, the dependence of the experimental SE_A_ on frequency was reduced.

Finally, the average theoretical SE_A_ of PVDF/MXene5 (6.13 dB) was close to the experimental value (4.57 dB), and other similar cases were PVDF/MXene2.5/ SWCNTs1 (the theoretical SE_A_ was 32.2 dB and the experimental value was 28.6 dB) and PVDF-SWCNTs1-MXene2.5 (the theoretical SE_A_ was 24.8 and the experimental value was 22.9 dB). The average theoretical SE_A_ of PVDF/SWCNTs2 (31.0 dB) was higher than the experimental value (19.4 dB). The formula assumes that the shield is an infinite conducting sheet illuminated by a plane wave and depends only on the frequency and on the conductivity, permeability, and thickness of the sheet [59]. Such an ideal shield does not exist in practical electronic system devices. In this study, the shield has an electrical discontinuity with the PVDF matrix. Understandably, the theoretical SE_A_ differed significantly from the experimental values. The average theoretical SE_A_ (21.2 dB) of PVDF-MXene2.5-SWCNTs1 was significantly lower than the experimental value (38.5 dB). The reason for this discrepancy may be that the theoretical and experimental SE_A_ calculations were different. It is known that the contribution of multiple reflection losses can be neglected in high-frequency electric fields when the skin depth is less than the shield thickness, or for materials with a total SE of more than 15 dB [32]. However, during the experimental process, in materials with numerous heterogeneous interfaces or pores, the incident EMWs lose significant energy as a result of multiple reflections due to interracial impedance mismatch, which was detected experimentally. Thus, the significance of multiple reflection losses cannot be disregarded when determining a heterogeneous material’s ability to absorb EMWs [32,58]. Notably, the analysis of planar shields can provide guidance for practical designs, and the theory remains reasonable under certain assumptions [60].

### 3.8. Electromagnetic Parameter Analysis

To better understand the electromagnetic absorption performance of the composites, the electromagnetic parameters, including relative complex permittivity (*ε*_r_ = *ε*′–j*ε*″) and relative complex permeability (*μ*_r_ = *μ*′–j*μ*″), were tested. *ε*′, *ε*″, *μ*′, and *μ*″ represent the storage and dissipation capabilities of electric and magnetic energy, respectively [61]. The dielectric loss tangents used to evaluate the dielectric loss capability (tan *δ_ε_* = *ε*″/*ε*′) [62,63] of the composite was also detected. Figure 7a shows the *ε*′ of PVDF/MXene5, PVDF/SWCNTs2, PVDF/MXene2.5/SWCNTs1, PVDF-SWCNTs1-MXene2.5, and PVDF-MXene2.5-SWCNTs1. The *ε*′ values of PVDF/SWCNTs1 (128.1–41.6) and PVDF/MXene2.5/SWCNTs1 (139.0–70.7) are higher than those of the composites, suggesting their higher energy storage capacity. The high conductivity, heterogeneous interfaces and abundant interfacial polarization of PVDF/SWCNTs1 and PVDF/MXene2.5/SWCNTs1 favor higher charge storage capacity. The *ε*′ of PVDF-MXene5 was only 7.50–7.60. Moreover, due to the dispersive behavior, *ε*′ and *ε*″ decreased with frequency [64]. Figure 7b presents the *ε*′ of PVDF/MXene5, PVDF/SWCNTs2, PVDF/MXene2.5/SWCNTs1, PVDF-SWCNTs1-MXene2.5, and PVDF-MXene2.5-SWCNTs1. The *ε*″ values of PVDF/SWCNTs1 (146.1–46.6) and PVDF/MXene2.5/SWCNTs1 (130.6–97.5) are also higher than those of other composites. The increase in *ε*″ value may have been caused by the increase in polarization and conduction losses, enhancing the attenuation of incident EM waves by interface polarization and relaxation [65].

Figure 7c shows the tan *δ_ε_* of PVDF/MXene5, PVDF/SWCNTs2, PVDF/MXene2.5/SWCNTs1, PVDF-SWCNTs1-MXene2.5, and PVDF-MXene2.5-SWCNTs1. The *ε*′ and *ε*″ of PVDF-MXene2.5-SWCNTs1 are not the largest among those of the composites, while PVDF-MXene2.5-SWCNTs1 achieved the highest tan *δ_ε_* (1.64–2.19), resulting in strong microwave dissipation [63] (The EMI SE_A_ of PVDF-MXene2.5-SWCNTs1 is as large as 39.4 dB), which is consistent with a moderate complex permittivity being more favorable for EM absorption performance [66]. The tan *δ_ε_* of the PVDF/SWCNTs1 composite reveals clear dielectric relaxation peaks with the EM frequency, indicating an additional loss due to dipole polarization [19]. Composites excluding PVDF/SWCNTs1 show small relaxation peaks, suggesting minor polarization loss. The dissipation of composite EM shielding is mainly caused by conductivity loss, which is also illustrated by their consistent SE_R_ (Figure 5c), absorption coefficient (Figure 5e), and tan *δ_ε_* (Figure 7c). Evidently, dielectric loss is the dominant contribution to microwave absorption in the composites. In addition, the permeability real (*μ*′) and imaginary (*μ*″) parts are less than 1 and close to zero, even taking negative values (Appendix A). The contribution to the EMI shielding can be considered negligible [46].

### 3.9. Mechanism Analysis

For polymer composites composed of heterogeneous fillers, according to the Maxwell Wagner theory [5], a large number of carriers accumulate at the interface of two materials, causing interface polarization in the mixed system with different dielectric permittivity. Due to the different dielectric permittivity and conductivities of MXene [6,64], SWCNTs [43,67], and PVDF [68], a large number of asymmetrically distributed charges accumulate at the interface of MXene, SWCNTs, and PVDF, which subsequently generate strong interfacial polarization. Simultaneously, the intrinsic dipole moments of molecules in the constituent medium align along the applied electric field direction, resulting in dipole polarization. The layered heterogeneous conductive filler with a cell-like structure of the segregated PVDF-MXene-SWCNTs composite structure repeatedly reflects and significantly attenuates the incident EM waves (Figure 8a,b) to obtain absorption-dominated efficient EM shielding, which occurs owing to the combined action of interface polarization, dipole polarization, and conductance loss caused by electron transition [69] (Figure 8c). The shielding mechanism of the EM waves in the PVDF-MXene-SWCNTs composite is illustrated in Figure 8.

The segregated PVDF composites are filled with layered fillers (SWCNTs and MXene). Composites filled with more SWCNTs exhibit increased absorption loss in induced conductivity. Simultaneously, impedance matching of the layered segregation structure was adjusted to improve the absorption loss to the total EMI SE, which synergistically improved the absorption-dominated EM shielding. Primarily, the synergistic effect of MXene and SWCNTs improves the impedance matching of the composite interface and reduces the reflectivity of EM waves. Second, SWCNTs located outside the structural unit cell are barrier-free lapped, ensuring excellent conductivity. Third, the numerous structural units inside the segregated structure reflect, attenuate, and absorb EM waves, thereby repeatedly increasing the EM shielding effect. In multiphase-interface composite materials, shielding effectiveness by multiple reflections is the dominant mechanism affecting the absorption capacity of EM waves [58]. Consequently, excellent impedance matching and dielectric loss promote multiple scattering, conductance loss, and dipole polarization, thereby considerably enhancing the absorption-dominated efficient EM shielding performance.

## 4. Conclusions

Composites of PVDF-MXene-SWCNTs with segregated structures have been successfully fabricated by electrostatic flocculation and hot pressing. The EM shielding of PVDF-MXene-SWCNTs is notably improved by the synergistic effect of excellent conductivity and impedance matching optimization due to the segregated layered structure. When the concentrations of MXene and SWCNTs are 2.5 wt% and 1 wt%, the dielectric loss (1.64–2.19) and impedance matching ratio (0.20–0.27) are higher than those of others. The conductivity of PVDF-MXene-SWCNTs composite is 2.75 S cm^−1^ and the EM SE reaches 46.1 dB with the EM shielding absorption coefficient of 85.7%. In layered segregated structures, the electromagnetic wave energy is attenuated by interfacial polarization, dipole polarization, and conduction losses. The innovative layered segregated structure ensures excellent conductivity through the barrier-free lapping of SWCNTs, thus obtaining efficient EMI shielding. The layered fillings of SWCNTs and MXene in the structure optimizes impedance matching and improves the absorption coefficient for electromagnetic shielding. A layered segregated structure with heterogeneous fillings provides a novel approach to simultaneously improve conductivity and impedance matching.

## Figures and Tables

**Figure 1 nanomaterials-13-00417-f001:**
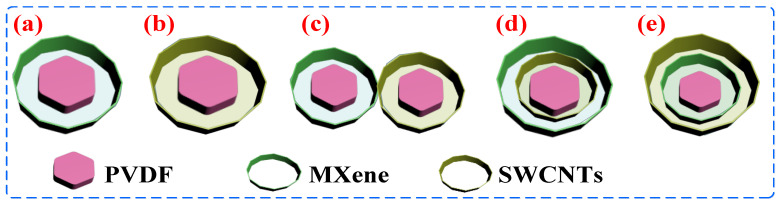
Schematic illustration of structure unit in (**a**) PVDF/MXene5, (**b**)PVDF/SWCNTs2, (**c**) PVDF/MXene/SWCNTs, (**d**) PVDF-SWCNTs-MXene, and (**e**) PVDF-MXene-SWCNTs composite.

**Figure 2 nanomaterials-13-00417-f002:**
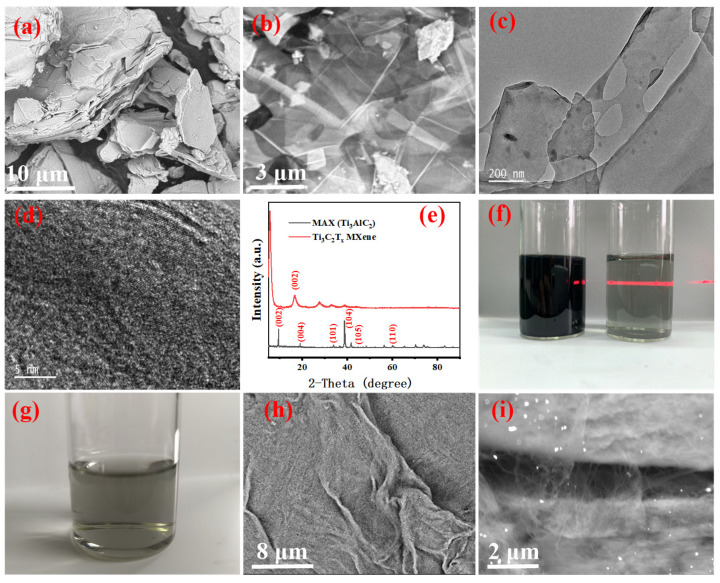
Scanning electron microscopy (SEM) images of MXene (**a**) before and (**b**) after ultrasound; (**c**) transmission electron microscopy (TEM) image of MXene; (**d**) high-resolution (HR)-TEM image of the flake edge of the MXene membrane; (**e**) X-ray diffraction (XRD) patterns of MAX and MXene; (**f**) Tyndall effect of MXene suspension before and after dilution; (**g**) photograph of diluted SWCNTs dispersed in the SDBS solution; SEM images of SWCNTs (**h**) before and (**i**) after dispersion.

**Figure 3 nanomaterials-13-00417-f003:**
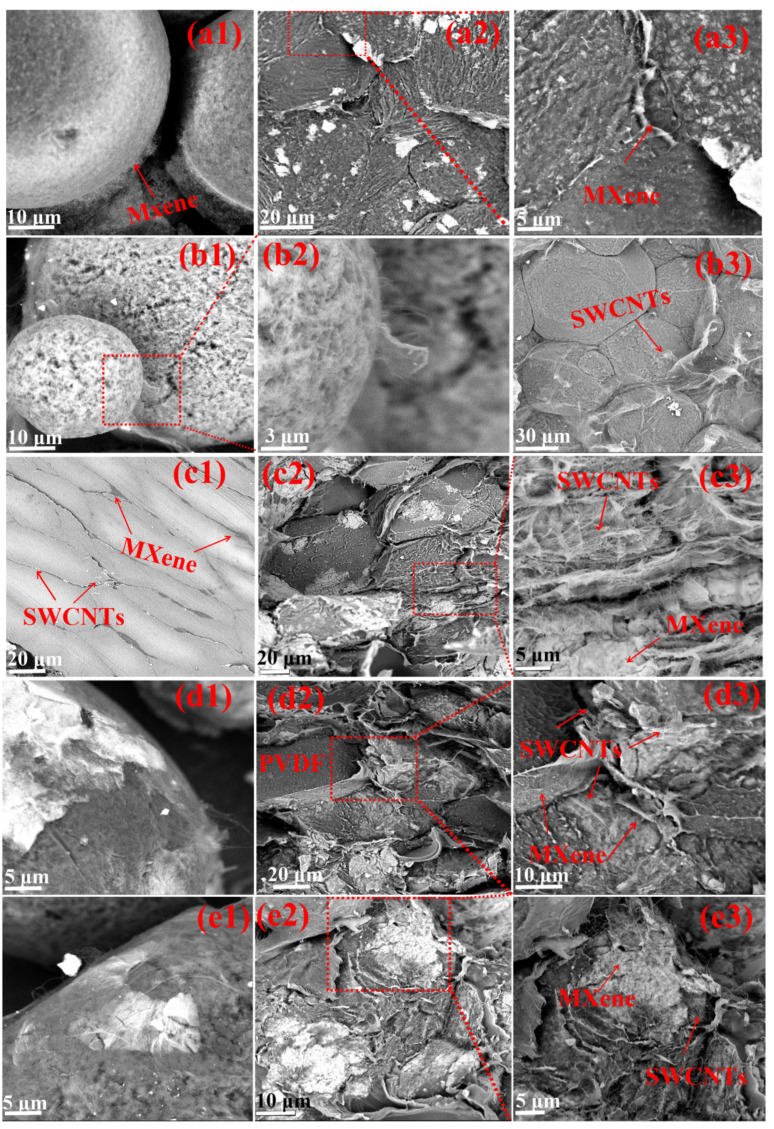
Morphology (**a1**) of PVDF/MXene5 microspheres; cross-sectional morphology (**a2**) and magnified view (**a3**) of PVDF/MXene composite. SEM (**b1**) and enlarged view (**b2**) of PVDF/SWCNTs2 microspheres; (**b3**) Cross-sectional morphology of PVDF/SWCNTs composite; (**c1**) cutting morphology, (**c2**) cross-sectional morphology, and (**c3**) the enlarged view of PVDF/MXene2.5/SWCNTs1. (**d1**) Morphology of PVDF-SWCNTs1-MXene2.5 microspheres; (**d2**) morphology and (**d3**) the enlarged view of PVDF-SWCNTs1-MXene2.5 composite. Morphology (**e1**) of PVDF-MXene2.5-SWCNTs1 microspheres; morphology (**e2**) and the enlarged view (**e3**) of PVDF-MXene2.5-SWCNTs1 composite.

**Figure 4 nanomaterials-13-00417-f004:**
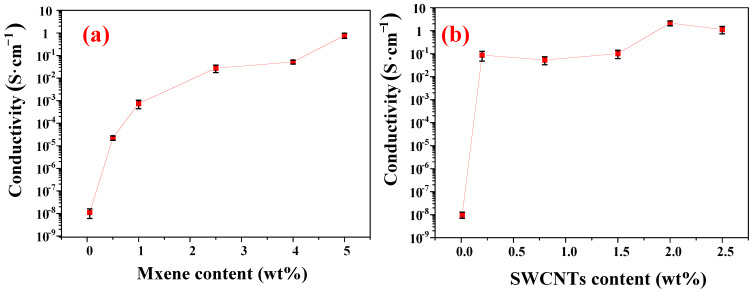
(**a**) Conductivity of PVDF/MXene composites versus MXene volume fraction; (**b**) conductivity of PVDF/SWCNTs composites versus SWCNTs volume fraction.

**Figure 5 nanomaterials-13-00417-f005:**
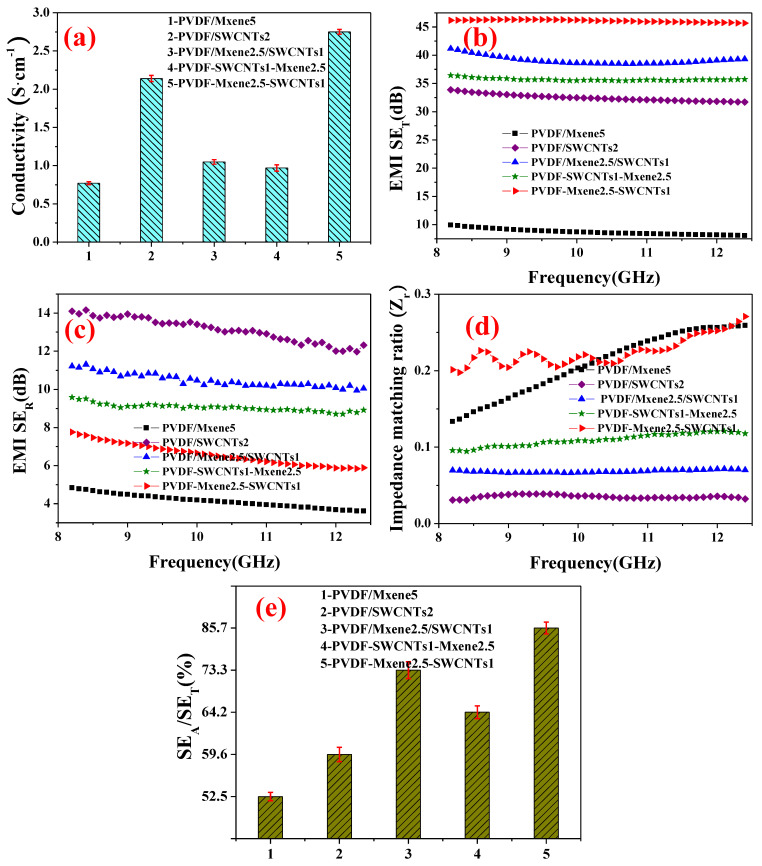
(**a**) Conductivity; (**b**) EMI SE_T_; (**c**) EMI SE_R_; (**d**) impedance matching ratio and (**e**) EM shielding absorption coefficient (SE_A_/SE_T_) of PVDF/MXene5, PVDF/SWCNTs2, PVDF/MXene2.5/SWCNTs1, PVDF-SWCNTs1-MXene2.5, and PVDF-MXene2.5-SWCNTs1.

**Figure 6 nanomaterials-13-00417-f006:**
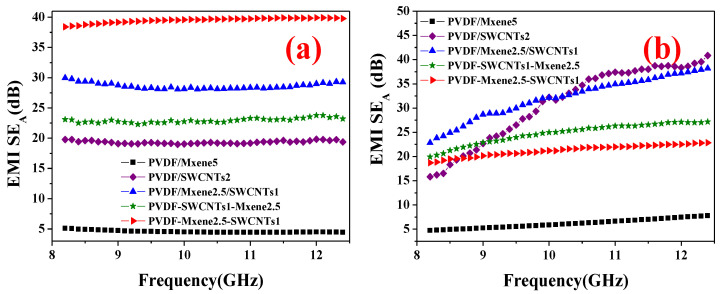
(**a**) Experimental and (**b**) theoretical EM shielding absorption (SE_A_) of PVDF/MXene5, PVDF/SWCNTs2, PVDF/MXene2.5/SWCNTs1, PVDF-SWCNTs1-MXene2.5, and PVDF-MXene2.5-SWCNTs1.

**Figure 7 nanomaterials-13-00417-f007:**
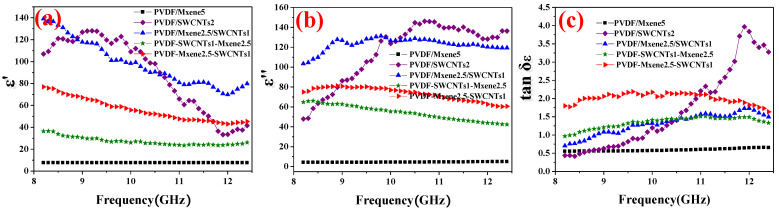
(**a**) *ε*′, (**b**) *ε″*, (**c**) dielectric loss (tan *δ_ε_* = *ε*″/*ε*′) of PVDF/MXene5, PVDF/SWCNTs2, PVDF/MXene2.5/SWCNTs1, PVDF-SWCNTs1-MXene2.5, and PVDF-MXene2.5-SWCNTs1.

**Figure 8 nanomaterials-13-00417-f008:**
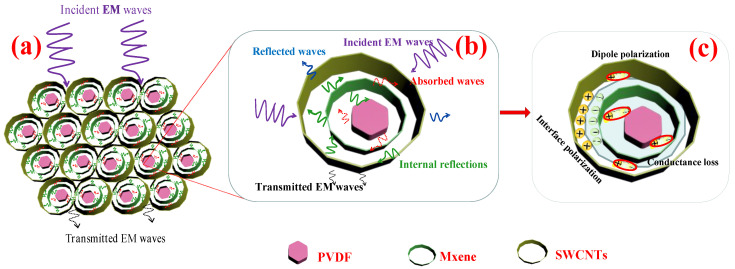
(**a**) EMI shielding mechanism in layered heterogeneous conductive-filler segregated structure; (**b**) repeated reflection and refraction of EM waves by segregated structure; (**c**) interfacial polarization, dipole polarization, and conduction loss of EM waves for PVDF-MXene-SWCNTs.

**Table 1 nanomaterials-13-00417-t001:** Weight by percent composition of the MXene/SWCNTs composite.

Samples	MXene Content (wt%)	SWCNTs Content (wt%)	Composition Structure Unit
PVDF/MXene5	5	0	PVDF/MXene5
PVDF/SWCNTs2	0	2	PVDF/SWCNTs2
PVDF/MXene2.5/SWCNTs1	2.5	1	PVDF/MXene5 and PVDF/SWCNTs2 ^1^
PVDF-SWCNTs1-MXene2.5	2.5	1	PVDF@SWCNTs1/MXene2.5
PVDF-MXene2.5-SWCNTs1	2.5	1	PVDF/MXene2.5@SWCNTs1

^1^ PVDF/MXene5 and PVDF/SWCNTs2 are the same mass.

**Table 2 nanomaterials-13-00417-t002:** Conductivity, EMI shielding, and SE_A_/SE_T_ according to the SWCNTs contents.

Samples *	SWCNTs Contents (wt %)	Conductivity (S cm^−1^)	EMI SE_A_ (dB)	EMI SE_T_ (dB)	SE_A_/SE_T_ (%)
PVDF/MXene2.5/SWCNTs0.2	0.2	2.89 × 10^−3^	12. 1	19.4	61.7
PVDF/MXene2.5/SWCNTs0.5	0.5	4.71 × 10^−1^	18.33	30.6	59.8
PVDF/MXene2.5/SWCNTs 1	1	1.05	30.2	41.2	73.3
PVDF/MXene2.5/SWCNTs 1.5	1.5	3.77	32.4	45.9	70.6

* MXene concentration in all the samples is 2.5%.

## Data Availability

The data presented in this study are available on request from the corresponding author.

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
