# Peer review of "Development of Electromagnetic-Wave-Shielding Polyvinylidene Fluoride–Ti3C2Tx MXene–Carbon Nanotube Composites by Improving Impedance Matching and Conductivity"

_nanomaterials, 2023, doi:10.3390/nano13030417_

Round 1
Reviewer 1 Report
In this paper, 'PVDF-MXene-SWCNTs' composite was fabricated by combining electrostatic and it showed improved EM shielding performance compare to 'PVDF/MXene/SWCNT'. The manuscript is well written. However, some issues remain unclear in this manuscript. There are some suggestions that need to be considered in revision.
1. Terminology is very confusing and some parts of the manuscripts used term incorrectly.
Ex) "Figure 1. Schematic illustration of the preparation of segregated PVDF/MXene/SWCNTs by electro-148 static assembly.". Maybe 'PVDF-MXene-SWCNTs' not PVDF/MXene/SWCNTs.
2. The author said, "The conflict between poor conductivity and optimal electromagnetic shielding has implications on the excellent impedance matching of PVDF/MXene2.5/SWCNTs 1. Therefore, 2.5% MXene and 1% SWCNTs were compared in this study." at line 301~303 of page 9.
Also, the author claims to have performed impedance matching optimization, but there is a lack of theoretical analysis and explanation for this.
Did the impedance matching optimization for the heterogeneous mixture of MXene and SWCNT be determined simply with the results of the four cases in Table 2? It is difficult to judge this as an optimization.
3. In Figure 5 (a) and (c). In general, since reflection occurs on the surface of a material, reflection occurs before EM absorption in the material. Therefore, as the conductivity of a material increases, it is common for 'EMI SER' to increase as well. However, in the case of '5-PVDF-Mxene2.5-SWCNTs1', conductivity increased significantly, but 'EM SER' due to reflection decreased rather. Why?
4. Theoretical EM absorbing performance can be calculated from the material's 'dielectric loss tangent', 'permittivity' and conductivity. How about presenting the permittivity in addition to the dielectric loss tangent and compare the theoretical EM wave absorption performance with experimental results?
Author Response
Dear reviewer:
We are grateful for your valuable advice and constructive comments on our manuscript. After carefully considering the recommendations, we have made several changes, and the manuscript has been improved as required. A certain revision is highlighted in green in the uploaded revised draft.​
Point 1:
Referee: Terminology is very confusing and some parts of the manuscripts used term incorrectly.
Ex) "Figure 1. Schematic illustration of the preparation of segregated PVDF/MXene/SWCNTs by electro-148 static assembly.". Maybe 'PVDF-MXene-SWCNTs' not PVDF/MXene/SWCNTs.
Response: We apologize for this oversight. We have rechecked the entire manuscript and modified the terminology based on your helpful comments to ensure that no similar problems are present.
Point 2:
Referee: The author said, "The conflict between poor conductivity and optimal electromagnetic shielding has implications on the excellent impedance matching of PVDF/MXene2.5/SWCNTs 1. Therefore, 2.5% MXene and 1% SWCNTs were compared in this study." at line 301~303 of page 9.
Also, the author claims to have performed impedance matching optimization, but there is a lack of theoretical analysis and explanation for this.
Response: We agree with you. The conductivity of PVDF/MXene2.5/SWCNTs1 (1.05 S cm-1) was intermediate, lying between those of PVDF/MXene5 (0.771 S cm-1) and PVDF/SWCNTs2 (2.14 S cm-1). Nevertheless, the EMI SET of PVDF/MXene2.5/SWCNTs1 was 41.2 dB, which was higher than those of PVDF/SWCNTs2 (33.9 dB) and PVDF/MXene5 (9.94 dB). The impedance matching ratios of composites were introduced to help understand the conflict between the unexceptional conductivity and prominent EM shielding. The phenomenon can be explained by the impedance matching ratios calculated for PVDF/MXene2.5/SWCNTs1 (0.066–0.072) higher than that of PVDF/SWCNTs2 (0.030–0.039) and better impedance matching achieve higher EMI shielding.
How the paper was modified: “We noted the contrast in the conductivity and electromagnetic shielding of the composite. The conductivity of PVDF/MXene2.5/SWCNTs1 (1.05 S cm-1) was between those of PVDF/MXene5 (0.771 S cm-1) and PVDF/SWCNTs2 (2.14 S cm-1). Nevertheless, the EMI SET of PVDF/MXene2.5/SWCNTs1 was 41.2 dB, which was higher than those of PVDF/SWCNTs2 (33.9 dB) and PVDF/MXene5 (9.94 dB). The contradiction between the unexceptional conductivity and prominent EM shielding is related to the better impedance matching of PVDF/MXene2.5/SWCNTs1 (0.066–0.072) than that of PVDF/SWCNTs2 (0.030–0.039).” The changes are highlighted in green in section “3.6. Conductivity, EMI Shielding, Impedance Matching, and Absorption Coefficient”
Referee: Did the impedance matching optimization for the heterogeneous mixture of MXene and SWCNT be determined simply with the results of the four cases in Table 2? It is difficult to judge this as an optimization.
Response: We thank you for pointing this out. It is difficult to judge the impedance matching optimization for heterogeneous mixtures of MXene and SWCNTs simply from the results of the cases in Table 2. In fact, Table 2 is used to compare the conductivity, EMI, and absorption coefficient (SEA/SET) according to the SWCNT concentrations. It is known that there is an opposite between optimized impedance matching and excellent outstanding conductivity [1, 2], both of which contribute to absorption-dominated EM shielding. To obtain efficient absorption-dominated EM shielding, 1% SWCNTs content is the appropriate SWCNT concentration for the optimal SEA/SET. As shown in Table 2, Figures 5d and e, the composite with 1% SWCNTs exhibits excellent EMI shielding with high absorption coefficient, including favorable impedance matching and outstanding conductivity.
Point 3:
Referee: In Figure 5 (a) and (c). In general, since reflection occurs on the surface of a material, reflection occurs before EM absorption in the material. Therefore, as the conductivity of a material increases, it is common for 'EMI SER' to increase as well. However, in the case of ' PVDF-Mxene2.5-SWCNTs1', conductivity increased significantly, but 'EM SER' due to reflection decreased rather. Why?
Response: We agree with your opinion. The reflection occurs on the surface during absorption within the materials [3]. Meanwhile, the superior impedance matching between the vacuum and shielding resulted in missing the reflection and enhancing the absorption [4].
In our study, PVDF-MXene2.5-SWCNTs1 (0.20–0.27) had the highest impedance matching ratio of composites except PVDF-MXene5, revealing the best absorption of EMI shielding (39.5 dB) and small reflection loss (6.56 dB). The conclusions are consistent with and verified by the present investigation [4,5]. The balance of impedance matching and electromagnetic loss eventually leads to low reflection coefficient and excellent microwave absorption performance [5].
How the paper is modified: “The best conductivity (2.75 S cm-1) (Figure 5a) and low average SER (6.56 dB) (Figure 5d) of PVDF-MXene2.5-SWCNTs1 are also attributed to the optimized impedance matching.”
“As shown in Figure 5d, PVDF-MXene2.5-SWCNTs1 (0.20–0.27) exhibited the highest impedance matching ratio among all composites except for PVDF-MXene5 (0.13–0.26), thereby revealing the highest absorption of EMI shielding (39.5 dB) and small reflection loss (6.56 dB). The superior impedance matching between the vacuum and shield resulted in an improved absorption value [48].” (These changes are highlighted in green highlighted in section 3.6.)
Point 4:
Referee: Theoretical EM absorbing performance can be calculated from the material's 'dielectric loss tangent', 'permittivity' and conductivity. How about presenting the permittivity in addition to the dielectric loss tangent and compare the theoretical EM wave absorption performance with experimental results?
Response: We appreciate this excellent good suggestion. Comparison of theoretical EM wave absorption performance with experimental results is shown and discussed in Section 3.7. The experimental and theoretical EM shielding absorption (SEA) are plotted in Fig. 6.
Permittivity and dielectric loss tangent are presented and analyzed in Section 3.8. Electromagnetic Parameter Analysis. The permittivity real (ε′) and imaginary parts (ε″) and dielectric loss (tan δε = ε″/ε′) of PVDF/MXene5, PVDF/SWCNTs2, PVDF/MXene2.5/SWCNTs1, PVDF-SWCNTs1-MXene2.5, and PVDF-MXene2.5-SWCNTs1 are shown in Figure 7.
For analytical completeness, the permeability real part (μ′) and imaginary parts (μ″) of PVDF/MXene5, PVDF/SWCNTs2, PVDF/MXene2.5/SWCNTs1, PVDF-SWCNTs1-MXene2.5, and PVDF-MXene2.5-SWCNTs1 are recorded in Figure S5.
How the paper was modified: “However, the overall trend is obviously different from the experimental SEA. First, the theoretical SEA was clearly different from the experimental SEA, which may be due to the assumptions in Eq. (4). Second, the dependence on frequency was different. The theoretical SEA was more dependent on frequency and relative electrical conductivity (σr). However, the experimental SEA was more complex, and the contribution from multiple reflections (SEM) cannot be ignored [57]. Thus, the dependence of the experimental SEA on frequency was reduced.
Finally, the average theoretical SEA of PVDF/MXene5 (6.13 dB) was close to the experimental value (4.57 dB), and other similar cases were PVDF/MXene2.5/ SWCNTs1 (the theoretical SEA was 32.2 dB and the experimental value was 28.6 dB) and PVDF-SWCNTs1-MXene2.5 (the theoretical SEA was 24.8 and the experimental value was 22.9 dB). The average theoretical SEA of PVDF/SWCNTs2 (31.0 dB) was higher than the experimental value (19.4 dB). The formula assumes that the shield is an infinite conducting sheet illuminated by a plane wave and depends only on the frequency and on the conductivity, permeability, and thickness of the sheet [58]. Such an ideal shield does not exist in practical electronic system devices. In this study, the shield has an electrical discontinuity with the PVDF matrix. Understandably, the theoretical SEA differed significantly from the experimental values. The average theoretical SEA (21.2 dB) of PVDF-MXene2.5-SWCNTs1 was significantly lower than the experimental value (38.5 dB). The reason for this discrepancy may be that the theoretical and experimental SEA calculations were different. It is known that the contribution of multiple reflection losses can be neglected in high-frequency electric fields when the skin depth is less than the shield thickness, or for materials with a total SE of more than 15 dB [32]. However, during the experimental process, in materials with numerous heterogeneous interfaces or pores, the incident EMWs lose significant energy as a result of multiple reflections due to interracial impedance mismatch, which was detected experimentally. Thus, the significance of multiple reflection losses cannot be disregarded when determining a heterogeneous material’s ability to absorb EMWs [32,57]. Notably, the analysis of planar shields can provide guidance for practical designs, and the theory remains reasonable under certain assumptions [59].”
Once again, we would like to take this opportunity to thank you for your decision and constructive comments on our manuscript.
References
[1] Thomassin, J.M.; Pagnoulle, C.; Bednarz, L, et al. Foams of polycaprolactone/MWNT nanocomposites for efficient EMI reduction[J]. J. Mater. Chem. 2008, 18 (7), 792–796.
[2] Su, X.; Wang, J.; Zhang, X., et al. Synthesis of core–shell Fe3O4@ ppy/graphite nanosheets composites with enhanced microwave absorption performance[J]. Mater. Lett. 2019, 239, 136–139.
[3] Raagulan, K.; Ghim, J.S.; Braveenth, R., et al. EMI shielding of the hydrophobic, flexible, lightweight carbonless nano-plate composites[J]. Nanomaterials 2020, 10(10), 2086.
[4] Magdi, S.; El-Diwany, F.; A Swillam, M. Broadband MIR harvester using silicon nanostructures[J]. Sci. Rep. 2019, 9(1), 1-7.
[5] Li W.; Guo F.; Zhao Y., et al. Facile Synthesis of Metal Oxide Decorated Carbonized Bamboo Fibers with Wideband Microwave Absorption[J]. ACS Omega, 2022, 7(43): 39019-39027.
Reviewer 2 Report
The authors present a somewhat interesting study of the MEI properties of PVDF microspheres coated with Mxene and/or SWNT.
The research is presented in a quite unclear way, and the writing is confusing in many places. Please try to improve the clarity of the whole text. In Figure 1 it would be helpful to include schematic representations (of the type shown in Figure 6) of the five different structures prepared/investigated. Please provide proper scale bars in all microscopy images.
Please point out more clearly what is the novelty of the authors approach. At the end of the introduction it is stated: "In this paper, we propose a strategy to improve the absorption coefficient of polymer-segregated structural composites considering the EM shielding effect."
Please explain the strategy expicitly so that the reader knows what to expect.
Two specific details:
Page 2, line 52; However, the agglomeration of MXenes results in a sharp reduction of their electrical conductivity, thus affecting their application in the fabrication of polymer matrices that exhibit electromagnetic shielding [10].
Is this information really available in reference 10? I could just find data of how etching time influenced resistivity, and that section ends with the statement "it is important to note that these results are preliminary and more work is needed before a good understanding of the effects of etching on transport properties is reached."
Page 6, line 216: Having these groups in the material can enhance its hydrophilicity, as well as make the surface electronegative.
Please rewrite in a more clear way.
In summary the manuscript is somewhat interesting, but the study is poorly presented.
Author Response
Dear reviewer:
Thank you for your decision and constructive comments on our manuscript. We have carefully considered the recommendations and made several changes.
We recognize that numerous parts of the manuscript need improvement based on your helpful comments. Corrections to reflect your comments are indicated by yellow highlight. The revised notes addressing the comments point-by-point are given as follows:
Point 1:
Referee: The research is presented in a quite unclear way, and the writing is confusing in many places. Please try to improve the clarity of the whole text.
Response: We have revised the entire manuscript. The revised manuscript has been edited and proofread by professional English-speaking editors at Editage.
Point 2:
Referee: In Figure 1 it would be helpful to include schematic representations (of the type shown in Figure 6) of the five different structures prepared/investigated.
Response: We agree with this advice. We have represented schemata of five different structural units in Figure 1 of the type shown in Figure 6. For clarity of expression, a schematic representation of the preparation of the five different structures is shown in Figure S1.
Point 3:
Referee: Please provide proper scale bars in all microscopy images.
Response: We agree with this advice. The scale bars have been provided in Figures 2 and 3, all microscopy images, to improve their clarity.
Point 4:
Referee: Please point out more clearly what is the novelty of the authors approach.
Response: The novelty lies in the prepared material, a layered-filler (SWCNTs and MXene) segregated structural composite exhibiting optimized impedance matching and excellent conductivity simultaneously. Based on your helpful comments, we have revised the original manuscript to highlight the novelty of our approach.​
How the paper was modified: We have added the sentence, “In this paper, we propose a novel strategy for improving conductivity and optimizing impedance matching by layered hetero-filling (MXene and SWCNTs) a PVDF-MXene-SWCNTs segregated structural composite.” at the end of the introduction in the manuscript (highlighted in yellow).
“The innovative layered segregated structure ensures excellent conductivity through the barrier-free lapping of SWCNTs, thus obtaining efficient EMI shielding. The SWCNTs and MXene layered filling of the structure optimizes impedance matching and improves the absorption coefficient for electromagnetic shielding.” at the end of the conclusions in the manuscript (Yellow highlighted).
Point 5:
Referee: At the end of the introduction it is stated: "In this paper, we propose a strategy to improve the absorption coefficient of polymer-segregated structural composites considering the EM shielding effect." Please explain the strategy expicitly so that the reader knows what to expect.
Response: We apologize for the unclear expressions in the text. We have modified and interpreted the strategy effectively in the following sentence.​
How the paper is modified: “In this paper, we propose a novel strategy for improving conductivity and optimizing impedance matching by layered hetero-filling (MXene and SWCNTs) of a PVDF-MXene-SWCNT segregated structural composite. The PVDF-MXene-SWCNTs segregated structural composite is fabricated by the electrostatic flocculation process and compression molding. SWCNTs and MXene are layered independently in the segregated PVDF composites optimized for impedance matching, leading to superior absorption coefficients. The layered fillings of SWCNTs and MXene in segregated PVDF composites allows their high conductivity to be preserved, resulting in excellent dielectric losses. The synergistic effect of a completely conducting network with a layered segregated structure consisting of layered SWCNTs and an MXene leads to excellent absorption-dominated EM shielding.” at the end of the introduction (highlighted in yellow).
Point 6:
Referee: Two specific details:
Page 2, line 52; However, the agglomeration of MXenes results in a sharp reduction of their electrical conductivity, thus affecting their application in the fabrication of polymer matrices that exhibit electromagnetic shielding [10].
Is this information really available in reference 10? I could just find data of how etching time influenced resistivity, and that section ends with the statement "it is important to note that these results are preliminary and more work is needed before a good understanding of the effects of etching on transport properties is reached."
Response: We apologize for this mistake. We checked the manuscript again and found that the literature is not relevant as per your comment. We have replaced [10] to ensure that the viewpoint of "exceptional accumulation of MXene layers results in a sharp reduction of their electrical conductivity" is based on cutting-edge research. Based on your helpful comments, we have rechecked all of the cited references to confirm those that are relevant to the research.
How the paper is modified: “Zhang S.; Ying H.; Yuan B.; Hu R.; Han W.-Q. Partial Atomic Tin Nanocomplex Pillared FewLayered Ti3C2Tx MXenes for Superior LithiumIon Storage. Nano-Micro Lett. 2020, 12, 78” (highlighted in yellow)
Referee: Page 6, line 216: Having these groups in the material can enhance its hydrophilicity, as well as make the surface electronegative. Please rewrite in a more clear way.
Response: This sentence has been corrected based on the helpful comments from the reviewers.
How the paper is modified: “The Ti atoms on the surface of the MXene easily bind to the molecules or ions in a solution, such as H2O and HF, and carry numerous groups, e.g., -F and -OH [39], which enhance hydrophilicity and make the surface electronegative (-106.78 mV) (Figure S3).” Highlighted in yellow in section 3.3.
Point 7:
Referee: In summary the manuscript is somewhat interesting, but the study is poorly presented.
Response: We apologize for the poor presentation of the previous version of the manuscript. Please rest assured that the manuscript has been thoroughly revised with the help of native English speakers to highlight the essence of the research and to improve its readability, organization, and credibility. We hope that, with this new version, our paper is now easier to follow and that language level has been substantially improved.
We would like to take this opportunity to thank you for your contribution to our manuscript. We would be happy to edit the text further based on the subsequent comments from the reviewers.​

Round 2
Reviewer 2 Report
The authors have made a comprehensive revsion and improved the clarity of the text and illustrations.